# Prenatal S-Adenosine Methionine (SAMe) Induces Changes in Gene Expression in the Brain of Newborn Mice That Are Prevented by Co-Administration of Valproic Acid (VPA)

**DOI:** 10.3390/ijms21082834

**Published:** 2020-04-18

**Authors:** Liza Weinstein-Fudim, Zivanit Ergaz, Moshe Szyf, Asher Ornoy

**Affiliations:** 1Department of Medical Neurobiology, Hebrew University Hadassah Medical School, Jerusalem 911200, Israel; Zivanit@hadassah.org.il; 2Department of Pharmacology and Therapeutics, McGill University Medical School, Montreal, QC H3A 2R7, Canada; moshe.szyf@gmail.com; 3Adelson School of Medicine, Ariel University, Ariel 4076405, Israel

**Keywords:** ASD, epigenetics, SAMe, VPA, gene expression, NanoString nCounter

## Abstract

In previous studies, we produced changes in gene expression in the brain of mice by early postnatal administration of valproic acid (VPA), with distinct differences between genders. The addition of S-adenosine methionine (SAMe) normalized the expression of most genes in both genders, while SAMe alone induced no changes. We treated pregnant dams with a single injection of VPA on day 12.5 of gestation, or with SAMe during gestational days 12–14, or by a combination of VPA and SAMe. In the frontal half of the brain, we studied the expression of 770 genes of the pathways involved in neurophysiology and neuropathology using the NanoString nCounter method. SAMe, but not VPA, induced statistically significant changes in the expression of many genes, with differences between genders. The expression of 112 genes was changed in both sexes, and another 170 genes were changed only in females and 31 only in males. About 30% of the genes were changed by more than 50%. One of the most important pathways changed by SAMe in both sexes was the VEGF (vascular endothelial growth factor) pathway. Pretreatment with VPA prevented almost all the changes in gene expression induced by SAMe. We conclude that large doses of SAMe, if administered prenatally, may induce significant epigenetic changes in the offspring. Hence, SAMe and possibly other methyl donors may be epigenetic teratogens.

## 1. Introduction

Autism spectrum disorder (ASD) is a neurobehavioral disorder manifested by persistent deficits in the two core domains of social interaction and communication, and fixed interests and repetitive behavior, along with developmental delay [1,2,3]. The prevalence of ASD, which is higher in males, is about 1% [3]. As of today, there are no biochemical or molecular markers that predict a diagnosis of ASD. Both genetics and environmental exposures, or potential interactions of the two, are postulated to cause ASD [1,2,3,4].

Valproic acid (VPA) is one of the most potent animal and human teratogens. Exposure to VPA during the first trimester of pregnancy is associated with significantly increased risks of major and minor congenital malformations, including neural tube defects (NTDs), limb defects, cardiovascular abnormalities, and cleft palate [5,6,7]. In animals, VPA causes dose-related teratogenic effects in all species investigated, including mice, rats, rabbits, and monkeys; these include skeletal malformations, and cardiac and craniofacial defects [8,9,10,11,12]. In rodents, exposure to a single dose of 500–600 mg of VPA on embryonic day 12, after the critical period of organogenesis, is associated mainly with an increased risk of autistic-like behavior [4,13,14,15].

Christianson et al. [16] first reported in 1994 that VPA intake during pregnancy leads to a significant increase in the rate of ASD in the offspring. Typical neurobehavioral symptoms of ASD can be detected in rodents exposed to VPA during pregnancy [13]. The association between VPA use in pregnancy and ASD, and the possibility to evaluate the behavioral changes by standard tests led to the development of experimental ASD models in rodents by VPA administered in high doses during different stages of pregnancy, or early postnatally [3,4,12,17].

The mechanism by which VPA causes ASD is not completely understood. Various studies showed that VPA may affect the neuronal system in different ways [18,19,20]. Kataoka et al. [18] showed that the exposure of mice to VPA at gestation day (GD) 12.5 caused a transient hyperacetylation of the histones H3 and H4 in the embryonic brain right after exposure, due to histone deacetylase inhibition. The offspring exhibited social interaction deficits, anxiety, and learning difficulties equivalent to findings among human ASD. Decreased neocortical neuronal density was observed in pathological studies.

VPA triggers increased oxidative stress, which may cause oxidative damage to lipids, DNA, and proteins [12,21,22,23]. Indeed, treatment of VPA-exposed dams with high doses of folic acid and vitamin E (antioxidants) and the methyl donor methionine ameliorated or prevented most VPA-induced damage [17,24].

The rapid increase in ASD rates in the last years has led to the hypothesis that ASD may also result from environmental perturbations that lead to epigenetic changes. Choi et al. [25] found that mice prenatally exposed to VPA who developed ASD in F1 also transferred some of the behavioral changes to the F2 and F3 generations, supporting the hypothesis that VPA induced epigenetic changes.

In pregnant rats, co administration of VPA (300 mg/kg/day on days 8, 9, and 10 of the pregnancy) and S-adenosyl methionine (SAMe) (10 mg/kg/day, on days 1–10 of the pregnancy) did not reduce the rate of fetal malformations associated with VPA exposure during pregnancy [26]. However, in our previous studies, we examined 60-day-old mice exposed to VPA on postnatal day 4 with co-administration of SAMe and found that SAMe significantly improved ASD-like behavior and reduced the brain’s oxidative stress induced by VPA [17]. Thus, treatment with SAMe can improve the behavioral abnormalities that result from the VPA-induced changes in the brain while it has no effect on the congenital anomalies. These and other findings indicated that VPA-induced autistic-like features accrue by different mechanisms than VPA-associated neural tube defects (NTDs) and other malformations. We also recently found that VPA induced in the brain changes in the expression of several genes associated with neurological development and inflammation; most of these changes were reversed by concomitant SAMe supplementation [27]. Postnatal administration of SAMe alone did not induce any changes in gene expression in the brain.

Post-translational histone modifications may modify gene expression regulation by controlling DNA accessibility to transcription factors. Tung et al. found that the exposure of dams to VPA at GD 9 resulted in a significant increase in histone acetylation in the offspring homogenate, which peaked 3 h after VPA exposure and was accompanied by an increase in histone methylation at histone H3 lysine 4 (H3K4) and a decrease in histone methylation at histone H3 lysine 9 (H3K9). Immunohistochemical staining performed on the embryos 3 h post exposure showed increased staining for the acetylated histone H3, particularly in the neuroepithelium and mesenchyme, as well as some increased staining in the heart and somites. VPA did not induce any significant changes in global or CpG island methylation at 1, 3, 6, and 24 h after VPA exposure, regardless of the presence of NTD at the 24-h time point [28].

Downregulation of several genes, including the methyl-CpG-binding protein 2 (MeCP2) gene, was found by Kim et al. among male rat offspring prenatally exposed to VPA [29].

S- adenosyl methinone (SAMe) is the principal methyl donor present in all living organisms and it is involved in multiple biochemical reactions. SAMe is critical for the regulation of cell growth, differentiation, function, and biosynthesis of hormones and neurotransmitters [30]. SAMe has also been shown to reduce oxidative stress [31,32,33,34]. Villalobos et al. [31] showed that SAMe modulates the cellular oxidative status, mainly by inhibiting lipid peroxidation and enhancing the glutathione system in the brain of a rat model of brain ischemia-reperfusion. Due to its important metabolic functions, especially in cellular trans-methylation pathways in the brain, SAMe is currently used as a food additive and as an adjunct treatment of several psychiatric and neurodegenerative diseases of the brain, such as depression, schizophrenia, and Alzheimer’s disease, with promising results [35,36]. The daily doses may range from 400–1200 mg (i.e., 8–20 mg/Kg body weight). Very few studies have examined the effect of SAMe treatment during pregnancy. Most of them focused on the efficacy of SAMe for the treatment of intrahepatic cholestasis of pregnancy (obstetric cholestasis), a common liver disorder specific to pregnancy, characterized by maternal pruritus and increased serum bile acid concentrations [37,38,39,40,41,42]. Most of these studies did not report on the neonatal outcome, except for preterm labor and a low Apgar score. One study carried out on an infant followed up to three months of age reported normal development [38]. Importantly, these clinical trials used intravenous injection of 800 to over 1000 mg/day of SAMe for a period between 14 and 20 days, during the third trimester of pregnancy [37,38,39,40,41,42], suggesting that SAMe is safe in the third trimester because no maternal or fetal adverse effects have been reported. On the other hand, the effectivity and the safety of SAMe administration during organogenesis were apparently not studied.

In the present study, we exposed the offspring of ICR mice to VPA on GD 12, with SAMe administered during days 12–14 of gestation or with a combination of VPA and SAMe. Our aim was to evaluate the offspring on post-natal day (PND)1 for the expression of genes related to physiology and pathology in the brain and to assess the gender specificity of these effects. We hypothesized that the VPA-exposed offspring, which could potentially develop ASD-like behavior, would exhibit increased brain oxidative stress and gene expression changes in the brain driven by epigenetic changes. We also hypothesized that due to the susceptibility of the embryo and fetus to epigenetic modulation, SAMe would induce significant changes in gene expression unrelated to VPA as well as interact with the effects of VPA on gene expression when combined with VPA. We hypothesize that these changes in gene expression underlie the mechanisms involved in VPA-triggered autism and its prevention with SAMe that we previously described.

We focused on the frontal half of the brain of PND1 mice because this part contains both the prefrontal cortex, which was found to be affected by postnatal VPA administration in our previous study, and the hippocampus, in which neuronal size abnormalities were found in ASD patients [43].

Furthermore, structural and genetic changes were also reported in these regions in mice with autistic-like behavior [44,45].

## 2. Results

There were no signs of toxicity or abnormal behavior in both VPA-treated and SAMe-treated dams, and all animals similarly gained weight during gestation. There was no significant effect of the VPA administration or of the administration of VPA plus SAMe on the litter average, or on the body weight of offspring. SAMe administration did not change the litter size (Table 1) but slightly and insignificantly reduced the weight of the offspring on PND1 (1.48g vs. 1.74 in the controls, *p* = 0.10). We did not find any external malformations in the offspring in any of the groups. The weight gain of the offspring during the first 30 postnatal days was similar in all groups (Figure 1A–C)

There were no differences in the developmental milestones between the control and SAMe-treated animals until the age of one month (Weinstein-Fudim L et al. [46]).

Similar to our previous findings of sex-specific VPA-induced differences in gene expression in adult mice [46], we found that newborn mice prenatally treated with SAMe demonstrated significant sex differences in the expression of genes in the brain, thus the data were analyzed for males and for females separately. 

The full list of the genes with significant changes in their expression is detailed in the Appendix A.

### 2.1. SAMe Administration

SAMe administration induced significant changes in the expression of many genes in both females and males. Changes in 112 genes overlapped in both sexes while many changes were sex specific. In females, SAMe significantly changed (*p* < 0.05) the expression of 170 additional genes, of which 56 of them were changed by more than 50% and 15 genes were changed by more than 100%. All genes that were significantly changed by 100% or more were upregulated (Table 2). In males, we had only two samples for the NanoString analysis of this group. However, since the data in both brains showed little variance, we decided to include them in the study. SAMe significantly changed the expression of 31 additional genes, 9 of which were changed by more than 50% but none were changed with more than 100%. Hence, they are not further described here (Figure 2).

Table 3 describes the genes whose expression was similarly changed in both genders. The expression of 112 genes was significantly changed in both genders, 25 genes changed by more than 50% with 23/25 changing in the same direction, 15 were upregulated, and 7 were downregulated. Eleven genes were upregulated by more than 100% in males and in females, except nostrin, which was changed by more than 100% in males and by 91% in females (Table 3).

### 2.2. Mouse Neonates Prenatally Treated with VPA and SAMe

Most of the genes that were changed by SAMe administration in males and in females were normalized to control levels by co-administration of single dose of VPA. Only four genes in males and one gene in females remained statistically significantly different from controls after treatment with VPA before SAMe administration (Table 4).

The expression of Prkcg (protein kinase C, gamma) and Vegf-*a* (vascular endothelial growth factor A) in males and Slc2a1 (solute carrier family 2 member 1) in females was significantly changed by SAMe administration. The co-administration of VPA did not affect the expression of Prkcg but downregulated the expression of Vegf-*a* and Slc2a1 to almost normal levels.

### 2.3. Comparison of Gene Expression Changes in Mouse Neonates Treated with VPA Versus those Treated with VPA + SAMe

In males, we found no changes in gene expression between the VPA and VPA + SAMe groups. In females, the expression of two genes was changed in the VPA+SAMe group in comparison to VPA. Axin2 (adj. *p* value= 0.01168), where the expression was also changed by SAMe alone and Desmin (Des, adj. *p* value= 0.05593). However, the levels of both genes were similar to their expression in the controls.

### 2.4. IPA Analysis

In males, the Ingenuity Pathway Analysis (IPA) analysis revealed a significant overexpression of genes associated with the VEGF pathway (Figure 3A,B). This pathway is proposed to have a significant role in tumor-derived VEGF-induced angiogenesis. In females, overexpression of the VEGF pathway was observed in fewer genes compared to males. VEGF angiogenesis was also overexpressed (in both genders). Specific genes changed in both genders are also detailed in Table 3.

The difference between Figure 3A,B is in the intensity of the colors, which show the level of the expression of a specific gene. For clarity, the data are shown in Table 5 (males) and Table 6 (females).

In males, the expression of 13 of 26 genes in the VEGF pathway was changed significantly after SAMe administration (pink), most of which were upregulated, as reported in Table 5 and Figure 3.

In females, only nine genes in this pathway were altered, as reported in Table 6.

In males, the z score of the VEGF pathway was statistically significant, meaning that the pathway was upregulated as predicted; in females, the pathway did not have a significant direction.

Anther pathway significantly activated in males with a positive Z score was the Interlukin8 signaling pathway. In females, the most significant canonical pathways are Cyclic adenosine monophosphate (cAMP)-mediated signaling and melatonin signaling pathways, with significant negative Z scores, predicting inhibited upstream regulators.

Figure 4. Heat maps in the different groups:

Figure 4A,B represent heat maps of the mRNA levels of the genes involved in neuropathological pathways in the frontal half of the brain. For each gene, the expression was normalized to the geometrical mean of seven housekeeping genes and the negative and positive technical controls. Each vertical column represents one animal belonging to the SAMe treatment group or control group indicated in the upper panel while each horizontal lane represents the normalized mRNA counts for one gene. The colors represent the expression of each gene among the different treated animals (red and blue represent strong and weak expression, respectively). As observed, there were many genes whose expression was significantly changed by SAMe in comparison to the controls. More of these genes were up- or downregulated in females compared to males.

Figure 5. Result of the MA plot showing a log two-fold change in the different genes studied in SAMe-treated versus control mice in both genders. More genes were up- or downregulated in females compared to males. In both genders, most of the genes whose expression was changed by SAMe were upregulated.

### 2.5. VPA Administration

Prenatal VPA exposure of the newborn mice did not induce any significant changes in gene expression of the frontal half of the brain (Figure 6A,B). As the treated mice were not followed to adulthood, since the gene expression of neonates obviously requires the sacrifice of animals, we could not determine whether these specific animals would have developed ASD-like behavioral changes. However, many studies have shown that prenatal administration of VPA to mice induced autistic-like behavior [4,13,14,15].

Figure 6 A,B show the effect of VPA administration during pregnancy on gene expression in female and male newborn mice.

Figure 7A,B show the effect of the combined VPA and SAMe administration during pregnancy on gene expression in female and male newborn mice.

## 3. Discussion

### 3.1. Absence of Gene Expression Changes in VPA-Exposed Newborn Mice

In our previous studies [46], we found behavioral abnormalities and multiple changes in gene expression in the prefrontal cortex of adult male and female mice exposed to VPA on postnatal day 4. This period is parallel to the third trimester of human pregnancy and a time of rapid maturation of the brain. We hypothesized that exposure to VPA during an earlier stage of pregnancy, and especially during the organogenesis period, will be manifested by substantial changes in gene expression in the neonates. In contrast to our hypothesis, we found no significant changes in gene expression following VPA injection during pregnancy.

Most studies that used the prenatal VPA model of ASD injected mice or rats with VPA during days 10.5–12.5 of pregnancy. Days 9–11 are the time of neural tube closure and organogenesis of brainstem cranial nerve nuclei cells. Rodents exposed to VPA during this period exhibit multiple anatomical abnormalities and behavioral features similar to those observed in autistic patients. We did not find neural tube or other external malformations since our VPA treatment was post neural tube closure and post active organogenesis of most organs.

Many studies explored changes in gene expression in this prenatal model of ASD. Most focused on specific genes or genes involved in processes of interest [47,48,49,50,51], and only a few studies have used comprehensive genetic analysis [14,52,53,54]. Huang et al. [52] used microarray gene expressing profiling analysis on hippocampus samples of 50-day-old male rats exposed to VPA on embryonic day 12.5. They detected 721 differently expressed genes, and most of them were found to be downregulated. Zhang et al. [53] performed whole transcriptome sequencing of the prefrontal cortex from 35-day-old male rats prenatally exposed to VPA. They reported 3228 differently expressed genes and 637 alternatively spliced genes between the VPA group and controls, among which were genes related to various neurological diseases, such as Huntington’s disease, Alzheimer’s disease, and Parkinson’s disease, and to pathways associated with neurogenesis.

Kotajima-Murakami et al. [14] found 2761 upregulated genes and 2883 downregulated genes in a whole mouse genome microarray analysis of brains from 11-week-old male mice prenatally exposed to VPA. Specifically, these mice exhibited an aberrant expression of genes associated with the mTOR (mammalian target of rapamycin) signaling pathway.

Hill et al. [54] explored changes in the methylation of CpG islands in the frontal cortex of 20-week-old male mice prenatally exposed to VPA and reported 11 differentially methylated regions associated with specific genes. They focused on the lower regulation and overexpression of only one of these genes, Chd7 (Chromodomain helicase DNA-binding protein 7), which is essential for neural crest cell migration. They did not focus on other changes in the methylation pattern, but the finding of specific methylation changes in adult mice exposed to VPA during pregnancy may indicate the involvement of epigenetic changes in key developmental genes, or that pathways that play a key role in the etiology of autism are affected.

In contrast to these studies examining large gene data, such as whole transcriptome sequencing or methylation analysis, which only sacrifice the rodents at adulthood, in the following behavioral analyses [14,52,53,54], we did not see changes in gene expression triggered by prenatal VPA at birth. It is possible that changes will be seen during later developmental stages, modulated via early epigenetic changes that trigger changes in expression later in life. Thus, it is possible that specific epigenetic changes induced by VPA during pregnancy will be reflected only at late-life time points. In addition, it is possible that the expression of other genes that are not included in the NanoString mouse Neuropathology Panel was changed in the brains of VPA-exposed pups.

### 3.2. Gene Expression Changes in SAMe-Exposed Newborn Mice

SAMe is the principal biological methyl donor in multiple methyltransferase reactions and a potent enhancer of DNA methylation, the most widely studied epigenetic modification, which influences chromatin structure and regulates gene expression. DNA methylation is crucial for gene expression regulation, fetal development, and brain function. In the human body, SAMe is synthesized directly from the amino acid methionine in an ATP-driven reaction [55].

DNA methylation plays an important role during embryonic development. Germ cell generation and early embryonic development are two developmental stages in embryonic development when most of the epigenetic reprogramming occurs, and the methylation pattern is dynamically changed. These periods are therefore vulnerable to environmental impacts, such as maternal nutrition. It is therefore not surprising that SAMe has substantial influence on gene expression when administered during the late embryonic and early fetal period (i.e., gestational days 12–14) while it has no effect following postnatal administration.

Significant reductions in SAMe levels is anticipated to contribute to DNA hypomethylation while increased levels induce DNA methylation and control gene expression [56]. However, in our study, most of the significantly changed genes were upregulated. These effects of SAMe might be indirect through the methylation and silencing of repressive sequences, miRNA, or repressor genes. Alternatively, SAMe affects the methylation of histones, such as H3K4 mono or trimethylation, which activates gene expression.

Our results demonstrated that SAMe supplementation during pregnancy can lead to massive changes in gene expression in the frontal half of the brain. This may have implications on the potential use of SAMe by pregnant mothers as well as to any potential adverse impact of alterations of the methyl content in the maternal diet. However, follow up on the offspring for one month showed normal growth and development and no mortality, implying that if there is some long-term damage, it is probably small. Additional studies are needed to further clarify this issue.

Although women are strongly recommended to take folic acid supplements before conception and during early pregnancy for the prevention of neural tube defects (NTDs) in offspring, high intake of folic acid and other one-carbon nutrients before and during pregnancy may be associated with both hypo- and hypermethylation in the offspring. For example, high maternal folic acid intake during pregnancy has been associated with reduced methylation at the insulin-like growth factor 2 (IGF2) differentially methylated region, H19 DMR, in cord blood DNA, especially in male offspring [57].

Periconceptional maternal folic acid intake was associated with the SAMe concentration in maternal serum and red blood cells but not with that of the child. However, the authors reported that it was related to increased methylation of IGF2 DMR of the child at 17 months of age [58]. Interestingly, higher IGF2 methylation in the child was also associated with lower birth weight.

Pauwels et al. [59] investigated the effects of maternal dietary methyl-group donor intake (choline, betaine, folate, and methionine) and supplemental intake of folic acid before and during each trimester of pregnancy on the methylation of several genes related to growth, metabolism, and appetite control in buccal epithelial cells of 6-month-old infants. They found that maternal methyl-group donor intake, especially during early gestation, induced epigenetic alterations in offspring genes related to metabolism and genes important to DNA methylation patterns.

Haggarty et al. [60] found that only late folic acid use, after 12 weeks of gestation, was associated with increased methylation levels of the IGF-2 gene and decreased methylation of PEG-3 (Paternally Expressed gene 3 ) and LINE-1 long interspersed nuclear element 1 (maternally methylated imprinted gene) in the cord blood [61]. Increased maternal serum vitamin B12 was associated with lower infant global DNA methylation [62].

Studies in rodents have shown that both maternal and paternal supplemental intake of methyl-group donors before conception and maternal intake during pregnancy can influence fetal methylation patterns, gene expression, metabolic status, and behavior in adult offspring [60]. The offspring of mice fed a methyl-supplemented diet during pregnancy have shown a phenotypic shift toward a brown coat color and altered feeding behavior [63]. The offspring of dams fed diets supplemented by high doses of folic acid during pregnancy displayed short-term memory impairment, decreased hippocampal size, and decreased thickness of the dentate nucleus at 3 weeks of age, and altered development of the cortical layers at embryonic day E17.5 [64].

Ryan et al. [65] reported that a methyl donor-enriched paternal diet (enriched for folic acid, L-methionine, choline, zinc, betaine, and vitamin B12), administered for 6 weeks before mating, impaired spatial learning and contextual fear conditioning, altered hippocampal gene expression, and impaired hippocampal synaptic plasticity in the F1 but not the F2 offspring generation. McCoy et al. [66] found that anxiety-prone male rats exhibited exacerbated anxiety-like and depression-like behaviors after dietary methyl donor depletion. When the methyl donor-depleted diet was given to the fathers for five weeks before mating, F1 offspring showed a similar behavioral phenotype in the open field and forced swim tests. These studies demonstrated the involvement of methyl donors in the epigenetic pattern changes in the offspring before and during pregnancy. It is important to note that studies examining the effect of methyl donor supplementation reported both hyper- and hypomethylation [67]. Consistent with these studies, our results demonstrated that SAMe increased or decreased gene expression in the brain during pregnancy. In our previous study, we detected no significant gene expression changes following SAMe exposure on postnatal days 4–6, possibly because the embryonic period is a rapidly proliferating and developing time with high vulnerability and nutrient demands to the brain, which slowly subsides postnatally. Although our study suggested that SAMe alone might have adverse effects, as mediated through extensive gene expression changes during the neonatal period, we showed that co-administration of VPA under conditions that are known to trigger ASD-like behaviors in offspring corrected most of these changes.

The fact that co-administration of VPA and SAMe abrogated these changes in gene expression supports the idea that they have overlapping targets, which is again consistent with SAMe overriding the ASD-like effects triggered by VPA. VPA and SAMe have opposite epigenetic effects. VPA increases histone acetylation [21] and DNA demethylation whereas SAMe is required for DNA methylation and histone methylation, which includes highly repressive marks, such as H3K9 methylation, which is antagonistic to histone acetylation [68,69], or H3K27 methylation [70]; both are highly repressive marks. Thus, VPA might counteract the epigenetic effects of SAMe, which is consistent with the potential of value of SAMe for the prevention of ASD.

Only a few studies have examined the influence of methyl donors on autistic-like behaviors in rodents. Langley et al. [71] found that high maternal choline consumption during pregnancy and lactation improved anxiety-like behaviors and increased social interaction in the BTBR T+tf/J (BTBR) mouse model of autism. Zhang et al. [72] found, using the same mouse model, that folic acid supplementation on postnatal days 14 to 35 reduced repetitive and stereotyped behavior, improved social communication, and enhanced memory and spatial learning. Furthermore, folic acid supplementation also reduced oxidative stress markers, neuronal loss, and decreased the levels of proinflammatory cytokines in the hippocampus. Huang et al. [73] examined the effects of treatment with betaine (an important methyl group donor and the substrate of betaine-homocysteine methyltransferase in methionine recycling and homocysteine metabolism) on offspring exposed to VPA during pregnancy. They reported that a single subcutaneous injection of betaine at 8 weeks of age reduced the homocysteine level in VPA-exposed mice, ameliorated social impairment and repetitive behavior, and normalized nociceptive sensitivity. We found no studies examining the effect of methyl donor treatment during pregnancy in the VPA model of ASD.

### 3.3. Vegf-a

Vascular endothelial growth factor (*Vegf-a*) is essential for embryonic vascularization and neural development. Interestingly, high serum levels of VEGF-a or its receptor FLT-1 (Fms Related Receptor Tyrosine Kinase 1) are associated with psychiatric and neurological alterations in adults [74,75,76].

In animal studies, knockout of the Vegf-*a* gene in mice, whether general or brain specific, was reported to be lethal [77]. Furthermore, even Vegf-*a* partial knockdown (heterozygous mice +/−) resulted in embryonic lethality due to multiple defects in vascular structure formation.

In contrast, adult mice overexpressing VEGF120, a short active variant of VEGF-a, in the forebrain exhibited profound differences in several forms of affective behaviors, such as reduced anxiety-like and fear-related behavior and reduced aggressive behavior. They may also be more resistant to a depression-like status [78].

### 3.4. Sex-Related Epigenetic Effects

In the last years, a number of studies have pointed to the possibility of sex-related effects of different agents, including hormones, drugs, and chemicals, as well as the characterization of gender-related diseases. For example, we found [79] that the effects of estradiol and testosterone on fetal rat bones in culture are sex specific. Testosterone increased bone formation only in fetal bones obtained from male fetuses and estradiol only affected female fetal bones. These gender differences in the response of bone and cartilage to hormones seems to be related to the effects of gonadal hormones, since several weeks after gonadectomy, these differences disappear.

Gender differences in the response to agents that may affect the epigenome were also described by several investigators. For example, maternal use of large amounts of alcohol during pregnancy may affect the developing embryo and fetus, inducing fetal alcohol spectrum disorder (FASD). One of the proposed mechanisms for this teratogenic action of ethanol is epigenetic modification and changes in the DNA methylation pattern. Indeed, several studies have shown that children born with FASD have methylation defects in genes related to glutamatergic synapses and protocadherins [80]. Similar changes were also observed following prenatal ethanol exposure in rodents [81]. Very recently, Amiri et al. found [82] that the DNA methylation of neural stem cells obtained from the brains of CD1 mouse fetuses is changed by chronic exposure to ethanol in vitro. The main changes were observed in genes related to glial markers. They also found that the pattern of changes was different between cells obtained from female fetuses compared to cells obtained from male fetuses.

Elsner et al. [83] found sex differences in DNA methyltransferase 1 content and histone deacetylase activity in the hippocampus of adolescent rats, both being higher in females compared to males. This points to gender-related differences in DNA methylation and histone acetylation activities. Whether this is an explanation of gender differences in the gene activity changes induced by methyl donors is as yet unknown. So far, we have no explanation for our current findings of sex differences in the effects of SAMe on fetal gene expression. Neither are we able to explain the prevention of these effects by a single concomitant dose of valproic acid.

### 3.5. Limitations of the Study

This study also has several limitations. First, we used only two male offspring that were prenatally exposed to SAMe. They were included because the data of both animals was very close. We also used doses of SAMe that were two to three times higher than the human dose, although the dams were treated for 3 days only as opposed to the human treatment, which is generally for a much longer time. In addition, we did not collect any data on gene silencing. Neither did we follow up the offspring to an age when we are able to carry out neurobehavioral studies and evaluate the autistic-like behavior induced by prenatal VPA and the possible damaging effects of SAMe. All these issues await additional studies in the future.

Our results demonstrate that the in utero developmental period serves as a critical time window during which maternal exposure to methyl donor supplementation can influence gene expression in the brain, as evidenced in the early postnatal period, with sex-specific differences, and may have a lifelong consequence on their health and behavior.

SAMe is a well-known food additive available in many stores without prescription and often recommended for use in pregnancy. The recommended SAMe dosage for humans is between 400 and 1200 mg/day [84,85,86], namely 8–15 mg/kg. Our dose was two to three times higher than the human dose. This is taken daily over long periods of time. We gave a higher dose but only for 3 days. Thus, our findings in mice should probably serve as a warning that SAMe has to be used very carefully in pregnancy. SAMe’s potential value in the treatment of various diseases should be weighed against the possible risk of adverse effects during pregnancy. Since we have shown SAMe has robust effects in the prevention of the development of ASD-like phenotypes postnatally, it is important to define a therapeutic window in the time and dose for SAMe for possible prevention and treatment of ASD.

## 4. Materials and Methods

### 4.1. Animals

On GD12, 12 ICR-CD1 pregnant female mice were injected subcutaneously with a single injection of either 600 mg/kg of valproic acid sodium salt (VPA; Sigma) in saline solution or with normal saline (NS). Each of these two groups (VPA and NS treated) were further subdivided into two groups: One group received 30mg/kg of SAMe (which is 2–3 times the human dose) by intraoral gavage once daily for 3 days starting on the day of VPA injection, and the other group similarly received phosphate-buffered saline.

On postnatal day 1 (PND 1), the pups were weighted and euthanized, and the frontal half of the brain was removed for the different studies.

The offspring of similarly treated mice were also followed until the age of one month with weighing and examination for the development of reflexes, and appearance of postnatal developmental milestones, including muscular strength, coordination, eye opening, surface righting, cliff aversion, rooting, forelimb grasp, auditory startle, ear twitch, and open field traversal tests as described by Hill et al. [87]. In this group, only physical sex determination was done; therefore, assessment in the first 2 weeks was on both genders as one group.

Animals were handled according to the NIH specifications with the approval of the committee for experimentation on animals of the Hebrew University Jerusalem, Israel.

Sex determination study for the newborn mice: We assessed the gender of each one of the day 1 pups by PCR using specimens obtained from their liver by the method described by McFarlene [88] and by us [89].

### 4.2. DNA Extraction for Sex Determination

Pieces of liver were placed in 100 μL of Extracta DNA Pref for PCR (Quanta biosciences), which allows rapid extraction of DNA that can be used directly in PCR reactions, eliminating the need for purification steps. Samples were heated at 95 °C for 30 min, cooled to room temperature, and dissolved by 100 μL of stabilizing buffer (safe stopping point). Then. 1.5 μL of each sample were added directly to a 25-μL PCR reaction.

#### PCR Reaction

To identify genes for the sex determination of newborn mice, we used two types of genes: The pseudo-autosomal genes Xlr and Sly (Xlr –x chromosome linked lymphocyte regulated complex, NM_011725 localized on the X and y chromosomes; Sly- Sycp3 -like Y-linked, BC049626 that regulates genes involved in chromatin remodeling and sperm differentiation), and Zfy, a gene on the Y chromosome that encodes a zinc-finger DNA binding protein. Each DNA extract was used in a PCR reaction with specific primer pairs for Sly/Xlr and Zfy genes, as stated below.

Primer sequences for the Xlr Sly and Zfy genes were obtained from McFarlane et al. (2013).

PCR Reactions: Genomic DNA was amplified with the following primer pairs:Sly/Xlr _F, 5ʹ-GATGATTTGAGTGGAAATGTGAGGTA-3ʹ; Sly/Xlr _R, 5ʹ-CTTATGTTTATAGGCATGCACCATGTA-3ʹ; Zfy_F, 5ʹ-GAC TAGACATGTCTTAACATCTGTCC-3ʹ; Zfy_R, 5ʹ-CC T A TTGC ATGGACTGCAGCTTATG-3ʹ

PCR reactions were performed in a final volume of 25 μL with 10µl AccuStart ll GelTrack PCR SuperMix that contained Taq DNA polymerase, Deoxynucleoside triphosphates (dNTPs) and electrophoresis tracking dyes (Quanta biosciences), 2 μL of forward primer, 2 μL of reverse primer (primers concentration: 10 pmol/µL), 9.5 µL of nuclease-free water, and 1.5 µL of sample, and the following PCR parameters: Initial denaturation at 95 °C for 5 min, 35 cycles with 95 °C for 30 s, 60 °C for 40 s, 72 °C for 30 s followed by final elongation at 72 °C for 5 min. PCR products were electrophoresed together with a DNA ladder (100 bp, Bio-lab) on 1% agarose gels and visualized with ethidium bromide under UV-illumination.

### 4.3. RNA Extraction and Gene Expression Analysis of the Brain

While looking at the upper surface of the cerebrum and cerebellum, we performed a coronal cut in the brain, exactly in the middle, removing the frontal half of the brain for our studies. The dissection was similar in all specimens (and was done by the same person, LW). Total RNA was extracted from the frontal half of the brains using the RNA/DNA/protein purification plus kit (47700; Norgen) according to the manufacturer’s protocol as described by us previously [15]. RNA was quantified using absorbance at 260 nm.

Gene expression analysis was performed on 18 samples, 4–5 pups in each group (excluding the saline+SAMe male group, with only 2 samples), using the NanoString nCounter system, which provides a simple way to profile specific nucleic acid molecules in a complex mixture. The system is based on direct digital detection of mRNA molecules utilizing target-specific color-coded probe pairs that can hybridize directly to target molecules. The expression level of mRNA molecules is measured by counting the number of times the barcode for that molecule is detected by a digital analyzer. It does not require the conversion of mRNA to cDNA by reverse transcription or the amplification of the resulting cDNA by PCR. The system does not need amplification and is sensitive enough to detect low-abundance molecules.

The data is expressed by the number of mRNA molecules in 100 ng/uL of RNA. It can simultaneously quantify up to 800 different interesting targets in a single reaction, making it ideal for miRNA profiling and targeted mRNA expression analysis [90]. We used the Mouse Neuropathology Panel, which includes 770 genes covering pathways involved in neurophysiology, neurodegeneration, and other nervous system diseases, and 10 internal reference genes for data normalization.

### 4.4. Statistical Analysis

NanoString analysis was performed on 3–5 samples from each group and each gender. In males, we had only two samples from the SAMe-treated group, hence, although reported, we note the limitation of the small sample.

Gene expression data were analyzed by the R package DESeq2, v1.22.1, Bioconductor [91]. Since samples were measured in two batches, the statistical model included both the treatment and the batch. After normalization by the internal reference genes, the Wald test was used to compare the different conditions, using default parameters, including the significance threshold of the Benjamini–Hochberg false discovery rate (FDR) (p adj) of less than 0.1. Further filtering of significant genes required a change in the expression of at least 50% relative to the control group.

Enriched canonical pathways of the significantly differentially expressed genes (FDR < 0.1) were identified using QIAGEN’s Ingenuity® Pathway Analysis (QIAGEN Inc., https://www.qiagenbioinformatics.com/products/ingenuity-pathway-analysis). All 770 genes of the Mouse Neuropathology Panel were taken as the background in the calculation. The scores were calculated by the right-tailed Fisher’s exact test. The scores derived from *p*-values indicate the likelihood of supplied genes belonging to a network versus those obtained by chance. A consistency score (Z-score) > 2 or < −2 indicates with ≥99% confidence that a supplied gene network was not generated by chance alone. Enrichment of “canonical pathways” and “up-stream regulators” with a Z-score > 2 or < −2 were considered for analysis.

Gene ontology for functional enrichment of pathways (KEGG) was performed for genes that were found to be significantly altered by VPA using DAVID bioinformatics resources 6.8. The total list of 770 genes related to neuropathology that were tested in the array were used as the background reference for the enrichment analysis. GO Biological Process (BP) enrichment was performed using the whole 770 genes of the nanostring neuropathology panel as a background (generated by running the function enrichGO in ClusterProfiler R package). No GO BP nor IPA canonical function passed Padj < 0.05.

## 5. Conclusions

SAMe, when administered to pregnant mice during the period of major organogenesis of the brain, induced very significant and sex-specific changes in gene expression in the frontal half of the offspring brain. This raises the possibility that SAMe is a neuroteratogen. A concomitant single injection of VPA to the dams abolished the epigenetic modulation action of SAMe, demonstrating that they have overlapping targets with antagonistic epigenetic effects. SAMe is often used as an adjunct treatment for a variety of psychiatric and neurological diseases and is freely sold as a food additive. Our data showed that SAMe should be used with caution in pregnant women. The mechanism underlying the modification of brain gene expression by SAMe, its sex-related effects, and the mode of the antagonistic effects of VPA have yet to be investigated.

## Figures and Tables

**Figure 1 ijms-21-02834-f001:**
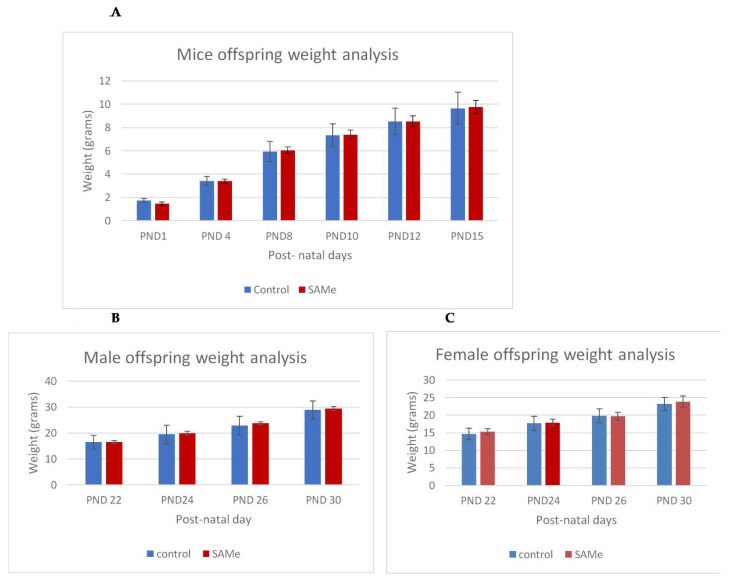
(**A**–**C**). Control versus S-adenosyl methionine (SAMe) offspring weight gain between post-natal day (PND)1 and PND30. Weight analysis of control versus SAMe pups between PND1 and PND15 days. Males and females combined. The number of pups in each group was 18. (**B**,**C**): Weight analysis of control vs. SAMe offspring between PND22 and PND30. (**B**) represents males and (**C**) represents females. Number of animals in each group: 6–9 in males, 9–12 in females.

**Figure 2 ijms-21-02834-f002:**
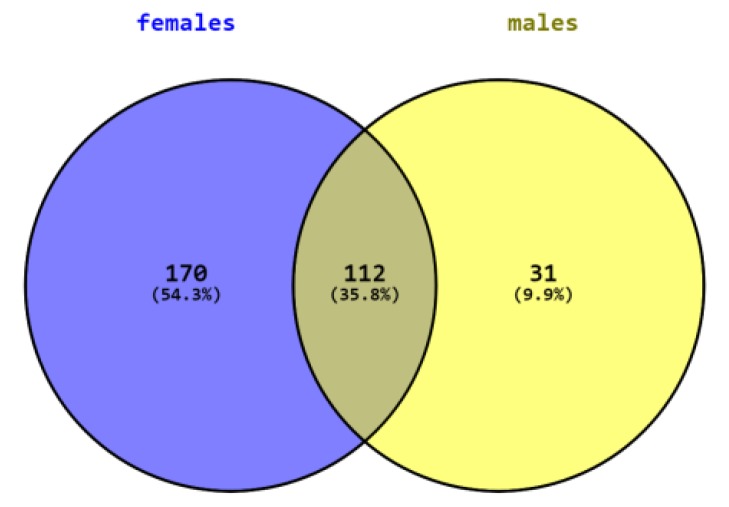
Number of genes differently expressed in males, females, and both sexes.

**Figure 3 ijms-21-02834-f003:**
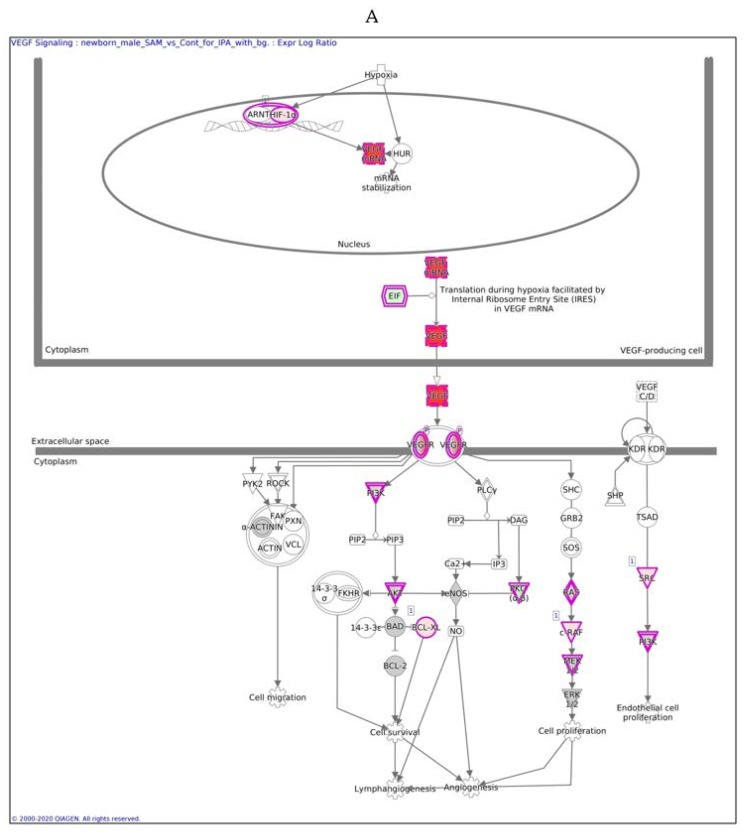
VEGF signaling pathway in males (**A**) and females (**B**) exposed to SAMe during pregnancy. Ingenuity pathway analysis of differentially expressed genes in mice treated with SAMe administration. Ingenuity Pathway (IPA) Analysis identified the VEGF-*a* (vascular endothelial growth factor A) signaling pathway as being enriched. The network was generated through the use of IPA (Qiagen, Ingenuity Systems, www.ingenuity.com) on normalized mRNA values measured by Nanostring analysis. Nodes represent molecules in a pathway, while the biological relationship between nodes is represented by a line (edge). Edges are supported by at least one reference in the Ingenuity Knowledge Base. The brightness of color in a node indicates the degree of upregulation (red) or downregulation (green). Nodes are displayed using shapes that represent the functional class of a gene product (Circle = Other, Nested Circle = Group or Complex, Rhombus = Peptidase, Square = Cytokine, Triangle = Kinase, Vertical ellipse = Transmembrane receptor). Edges are marked with symbols to represent the relationship between nodes (Line only = Binding only, Flat line = inhibits, Solid arrow = Acts on, Solid arrow with flat line = inhibits and acts on, Open circle = leads to, Open arrow = translocates to).

**Figure 4 ijms-21-02834-f004:**
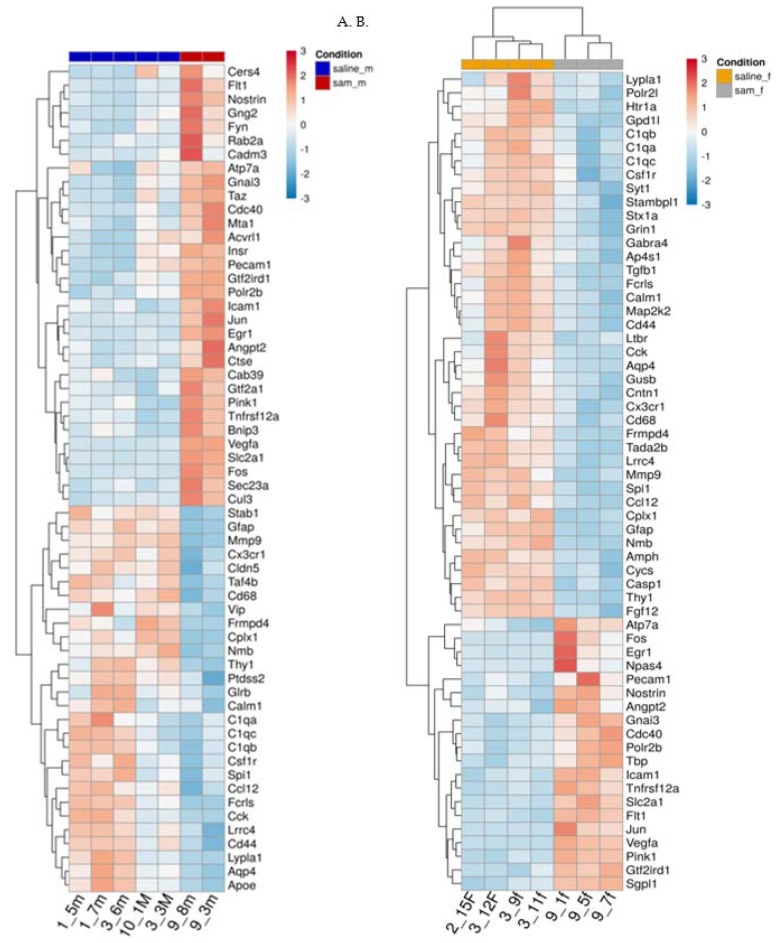
(**A**,**B**). Effect of Valproic acid (VPA) and SAMe administration on gene expression in the frontal half of the brain. Heat map of the genes significantly changed by SAMe in comparison to controls. A. left column - females, B. right column - males.

**Figure 5 ijms-21-02834-f005:**
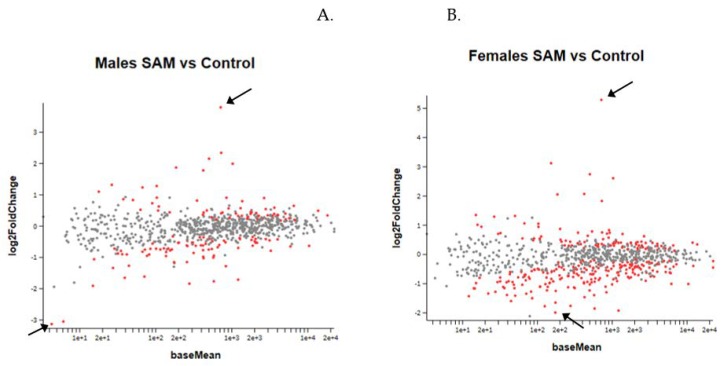
MA Plot of log 2 fold-change (*y*-axis) versus base mean (normalized average expression (*x*-axis). Every dot represents one gene. Red dots: genes with a statistically significant change compared to controls. Grey dots: no significant change. Higher and lower 2-fold change red dots are marked by arrows. (**A**). Gene expression in males. (**B**). Gene expression in females.

**Figure 6 ijms-21-02834-f006:**
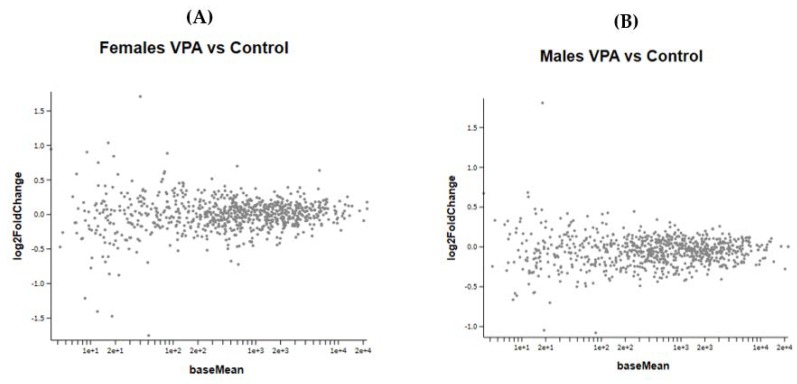
(**A**,**B**) show the MA plot of a two-fold log change in the different genes studied in VPA-treated versus control mice in both genders No statistically significant changes were observed between the VPA and control groups in both male and female newborn mice.

**Figure 7 ijms-21-02834-f007:**
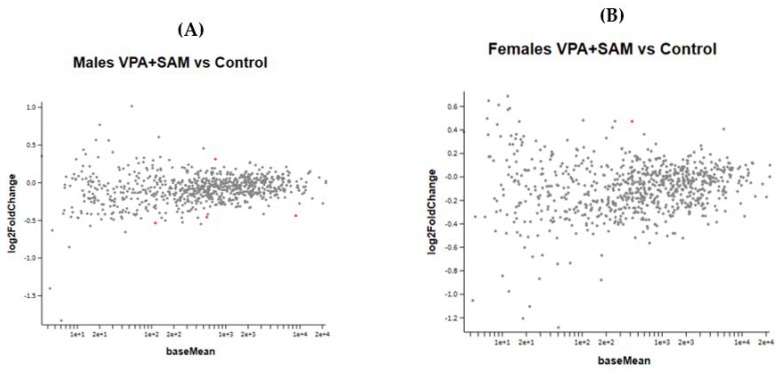
(**A**,**B**) show the MA plot of a two-fold log change in the different genes studied in VPA+ SAMe-treated versus control mice in both genders. In females, only Slc2a1 (solute carrier family 2 member 1) was found to be changed in the VPA+ SAMe group. This gene changed by 445% by SAMe administration alone compared to the control and was reduced by VPA injection to 40% of the change (Table 7). In males, four genes were changed by co administration of VPA+ SAMe: Fus (fused in sarcoma), Pdgfr (platelet derived growth factor receptor, beta polypeptide), Prkcg (protein kinase C, gamma), and Vegfa (vascular endothelial growth factor A). In Vegfa, VPA administration corrected the upregulation caused by SAMe from 390% to 28%, but the change was still statistically significant. In the other three genes, VPA injection did not change, or slightly increased, the effect of SAMe alone, as reported in Table 7.

**Table 1 ijms-21-02834-t001:** Litter size analysis of the different treatment groups.

		Control	SAMe	VPA	VPA+SAMe
litter size	average	11	10.75	11	12.33
	s. dev	2.449	2.168	2	1.528
	*p* value SAMe-control	0.209			
	*p* value VPA-control	0.355			

**Table 2 ijms-21-02834-t002:** Genes significantly changed at least by 100% in females after SAMe administration.

Symbol	Official Full Name	Log2 Fold Change	Ajd *p* Value	% Change	Neuroinflammation	Neuroplasticity, Development & Aging	Metabolism	Compartmentalization and Structural Integrity	Neuron-Glia Interaction	Neurotransmission	
Npas4	neuronal PAS domain protein 4	3.121	0.000	479%	+	-	-	+	-	+	
Tbp	TATA box binding protein	0.872	0.000	134%	-	-	+	-	-	-	
Pdgfrb	platelet derived growth factor receptor, beta polypeptide	0.614	0.002	176%	+	+	+	-	-	+	
Ptgs2	prostaglandin-endoperoxide synthase 2	1.35	0.002	763%	+	+	+	-	-	+	reported in human and animal Sfari data base to be related to ASD
Msn	moesin	0.505	0.003	143%	-	-	-	+	-	-	reported in human Sfari data base to be related to ASD
Igf1	insulin-like growth factor 1	−1.024	0.007	254%	+	+	+	-	-	+	
Mmp14	matrix metallopeptidase 14 (membrane-inserted)	0.401	0.009	199%	+	+	+	-	-	-	
Nfkbia	nuclear factor of kappa light polypeptide gene enhancer in B cells inhibitor, alpha	0.471	0.012	112%	-	+	-	-	+	+	
Smyd1	SET and MYND domain containing 1	−0.920	0.030	877%	-	+	-	-	-	-	
Erbb3	erb-b2 receptor tyrosine kinase 3	−1.41	0.033	372%	-	+	-	-	-	+	
Pmp22	peripheral myelin protein 22	0.438	0.036	400%	-	-	-	-	+	+	
Napsa	napsin A aspartic peptidase	1.016	0.040	709%	-	-	+	-	-	-	
Tnfrsf10b	tumor necrosis factor receptor superfamily, member 10b	0.732	0.066	409%	+	+	-	-	-	-	
Kcna1	potassium voltage-gated channel, shaker-related subfamily, member 1	−0.527	0.092	115%	-	-	-	+	-	+	
Cdk2	cyclin-dependent kinase 2	0.310	0.098	179%	-	+	+	-	-	-	reported in animal Sfari data base to be related to ASD

+ and - are related to the belonging of the genes to a fundamental theme of neurodegeneration.

**Table 3 ijms-21-02834-t003:** Genes changed at least by 100% in both genders after SAMe administration alone.

Gene	Official Full Name	Adjusted *p* Value Females	% Change- Females	Adjusted *p* Value Males	% Change-Males	Neuroinflammation	Neuroplasticity, Development & Aging	Metabolism	Compartmentalization and Structural Integrity	Neuron-Glia Interaction	Neurotransmission	
Vegfa	vascular endothelial growth factor A	30,000	292%	0.000	390%	+	+	-	-	-	-	
Slc2a1	solute carrier family 2 (facilitated glucose transporter), member 1	0.000	445%	0.000	263%	+	-	-	+	-	-	
Jun	jun proto-oncogene	10,000	449%	0.000	317%	-	+	+	-	+	+	
Fos	FBJ osteosarcoma oncogene	0.000	4294%	0.000	1516%	-	+	+	-	-	+	
Flt1	FMS-like tyrosine kinase 1	0.000	272%	0.000	219%	+	+	-	-	-	-	reported in human sfari data base to be related to ASD
Egr1	early growth response 1	0.000	664%	0.000	375%	-	+	-	-	-	-	
Icam1	intercellular adhesion molecule 1	0.000	415%	0.013	110%	-	-	-	+	-	-	
Nostrin	nitric oxide synthase trafficker	0.000	91%	0.000	138%	-	-	-	-	-	-	
Tnfrsf12a	tumor necrosis factor receptor superfamily, member 12a	0.000	372%	0.000	218%	+	+	-	-	-	-	
Ctse	cathepsin E	0.006	233%	0.015	226%	+	-	+	-	-	-	
Bnip3	BCL2/adenovirus E1B interacting protein 3	0.013	187%	0.0015	108%	-	-	+	+	-	-	
Cldn15	claudin 15	0.028	239%	0.024	213%	-	-	-	+	-	-	
Nostrin	nitric oxide synthase trafficker	0.000	91%	0.000	138%	+	-	-	-	-	-	

+ and - are related to the belonging of the genes to a fundamental themes of neurodegeneration.

**Table 4 ijms-21-02834-t004:** Genes that were still changed by SAMe in spite of the prior administration of VPA compared to controls in males and/or females.

	Gene	Official Full Name	*p* Value	Adjusted *p* Value	% Change Versus Control	% Change Versus SAME Alone	Neuroinflammation	Neuroplasticity, Development & Aging	Metabolism	Compartmentalization and Structural Integrity	Neuron-Glia Interaction	Neurotransmission
Males	Fus	fused in sarcoma	0.000	0.005	−35%	−21%	-	-	-	+	-	-
	Pdgfrb	platelet derived growth factor receptor, beta polypeptide	0.000	0.051	−44%	−7%	+	+	+	-	-	+
	Prkcg	protein kinase C, gamma	0.000	0.086	−35%	−32%	-	+	-	+	-	+
	Vegfa	vascular endothelial growth factor A	0.000	0.086	30%	424%	+	+	-	-	-	-
Females	Slc2a1	solute carrier family 2 (facilitated glucose transporter), member 1	0.000	0.098	40%	445%	+	-	-	+	-	-

*The expression of Prkcg and Vegfa in males, and Slc2a1 in females was also significantly changed by SAMe alone. + and - are related to the belonging of the genes to a fundamental theme of neurodegeneration.

**Table 5 ijms-21-02834-t005:** Changes in the expression of genes in the vascular endothelial growth factor (VEGF) pathway in males induced by SAMe.

Symbol	Gene Name	Expr Intensity/RPKM/FPKM/Counts	Expression Log Ratio	Expression p-value	Expr False Discovery Rate (q-value)	Change Percent	Expected Change	Gene Direction	Location
AKT3	AKT serinethreonine kinase 3	5928.448	0.346	0.007	0.055	21%	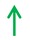	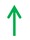	Cytoplasm
BCL2L1	BCL2 like 1	1259.759	0.222	0.009	0.066	17%	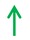	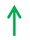	Cytoplasm
EIF2S1	eukaryotic translation initiation factor 2 subunit alpha	2055.588	−0.407	0.006	0.049	−17%		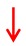	Cytoplasm
FLT1	fms related tyrosine kinase 1	185.192	1.872	0.000	0.000	219%	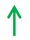	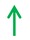	Plasma Membrane
HIF1A	hypoxia inducible factor 1 subunit alpha	4400.794	0.181	0.009	0.066	13%	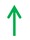	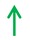	Nucleus
HRAS	HRas proto-oncogene, GTPase	4003.534	−0.421	0.013	0.086	−16%	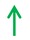	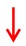	Plasma Membrane
MAP2K2	mitogen-activated protein kinase kinase 2	2299.153	−0.219	0.014	0.090	−12%	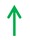	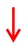	Cytoplasm
PIK3CA	phosphatidylinositol-4,5-bisphosphate 3-kinase catalytic subunit alpha	1824.24	0.442	0.005	0.046	23%	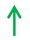	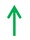	Cytoplasm
PRKCA	protein kinase C alpha	826.876	-0.515	0.006	0.051	-38%	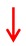	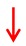	Cytoplasm
RAF1	Raf-1 proto-oncogene, serinethreonine kinase	2563.201	0.244	0.012	0.078	20%		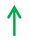	Cytoplasm
RRAS	RAS related	107.351	0.499	0.0128	0.082	52%	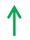	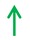	Cytoplasm
SRC	SRC proto-oncogene, non-receptor tyrosine kinase	2265.583	0.331	0.0174	0.096	14%	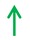	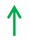	Cytoplasm
VEGFA	vascular endothelial growth factor A	725.666	2.343	0.000	0.000	390%	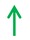	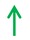	Extracellular Space

The “expected change” column indicates the state that the gene is predicted to have if the pathway was activated. In the “gene direction” column we can see if the gene is acting as expected, as also seen in the “expression log ratio” column, which represents the Log2 fold change of the gene. Green arrows- upregulation of the gene. Red arrows-downregulation of the gene.

**Table 6 ijms-21-02834-t006:** Changes in the expression of genes in the VEGF pathway in females induced by SAMe.

Symbol.	Gene Name	Expr Intensity/RPKM/FPKM/Counts	Expr Log Ratio	Expr *p*-Value	Expr False Discovery Rate (q-Value)	Change Percent	Expected	Gene Direction	Location
EIF2S1	eukaryotic translation initiation factor 2 subunit alpha	2056	−0.292	0.032	0.093			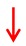	Cytoplasm
FLT1	fms related tyrosine kinase 1	185.2	2.053	0.000	0.000	272%	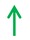	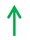	Plasma Membrane
HRAS	HRas proto-oncogene, GTPase	4004	−0.588	0.000	0.002	−25%	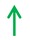	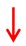	Plasma Membrane
MAP2K2	mitogen-activated protein kinase kinase 2	2299	−0.452	0.000	0.000	−26%	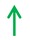	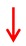	Cytoplasm
PRKCA	protein kinase C alpha	826.9	-0.542	0.002	0.011	−51%	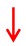	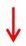	Cytoplasm
RAF1	Raf-1 proto-oncogene, serinethreonine kinase	2563	0.247	0.006	0.027	17%	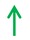	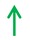	Cytoplasm
RRAS	RAS related	107.4	0.543	0.003	0.020	357%	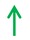	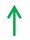	Cytoplasm
SRC	SRC proto-oncogene, non-receptor tyrosine kinase	2266	0.306	0.017	0.059	−5%	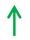	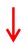	Cytoplasm
VEGFA	vascular endothelial growth factor A	725.7	1.829	0.000	0.000	292%	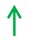	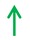	Extracellular Space

Green arrows- upregulation of the gene. Red arrows-downregulation of the gene.

**Table 7 ijms-21-02834-t007:** Genes significantly changed in males and females by co-administration of SAMe and VPA.

	Gene	Official Full Name	*p* Value	Adjusted *p* Value	% Change Versus Control	% Change Control versus SAME	Neuroinflammation	Neuroplasticity, Development & Aging	Metabolism	Compartmentalization and Structural Integrity	Neuron-Glia Interaction	Neurotransmission
	Fus	fused in sarcoma	0.000	0.005	−24%	−18%	-	-	-	+	-	-
Males	Pdgfrb	platelet derived growth factor receptor, beta polypeptide	0.000	0.051	−29%	−6%	+	+	+	-	-	+
	Prkcg	protein kinase C, gamma	0.000	0.086	−24%	−38%	-	+	-	+	-	+
	Vegfa	vascular endothelial growth factor A	0.000	0.086	28%	390%	+	+	-	-	-	-
Females	Slc2a1	solute carrier family 2 (facilitated glucose transporter), member 1	0.000	0.098	40%	445%	+	-	-	+	-	-

+ and - are related to the belonging of the genes to a fundamental theme of neurodegeneration.

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
