# Peer review of "Prenatal S-Adenosine Methionine (SAMe) Induces Changes in Gene Expression in the Brain of Newborn Mice That Are Prevented by Co-Administration of Valproic Acid (VPA)"

_ijms, 2020, doi:10.3390/ijms21082834_

Round 1

Reviewer 1 Report

This is an interesting comparison between VPA and SAMe treatment for potential sexual dimorphic response at genetic level. Using Nanostring is a relatively quick and easy to interpret ways to have a first look.

Could you clarify if VPA has no changes in both male and female, are there data?

Author Response

Response to reviewer's comments:

We thank the reviewers for their instructive comments. Here is our response.

Reviewer 1:

We thank this reviewer for his Comment. VPA is a well -known teratogen and neuro-teratogen in human as well as in animals, as written in the introduction. It also serves as an important model for the induction of autistic like behavior in rodents. In the present study we administered only a single dose of VPA to the dams, on day 12, which is after the usual time for teratogenic effects but within the time for its neuro-teratogenic effects and for the induction of ASD like behavior. This was now added to the introduction. We also added a sentence regarding the teratogenicity of VPA in the introduction and further elaborated on this issue in the second paragraph of the discussion.

Could you clarify if VPA has no changes in both male and female, are there data?

The data on gene expression of VPA treated compared to controls were added as figure 6 and in the text of the results section and the discussion.

Reviewer 2 Report

In the present manuscript by Weinstein-Fudim et al. entitled:

‘Prenatal S-adenosine methionine (SAMe) induces changes in gene expression in the brain of newborn mice which are prevented by co-administration of valproic acid (VPA)’

authors analyzed the effect of valproic acid (VPA) and S-adenosine methionine (SAMe) administered to pregnant dams in the frontal part of brain of the offspring assessing the differential gene expression of a panel of 770 genes involved in neurophysiology by using NanoString nCounter. They observed that VPA counteracted almost all sex-specific changes in gene expression induced by SAMe concluding that SAMe administered prenatally may induce significant epigenetic changes.     

After an exhaustive analysis of the manuscript by Weinstein-Fudim et al., I did the following observations:

  1. It is a document that contains some interesting elements such as showing that SAMe, an antioxidant and methyl donor commonly administered as adjunct treatment in a variety of neurological diseases, modified gene expression in the brain of pups exposed prenatally at doses 3 times higher than the recommended in human (according to authors), and in these conditions VPA had an antagonist effect. Thus, authors suggest that SAMe should be used with caution in pregnant woman. This information may be important after several studies showing that prenatal exposure to VPA leads to increased rates of autism and in mice SAMe might counteract these effects, which may tempt to an indiscriminate (and not fully demonstrated) use of SAMe to prevent autism.

However, this information may be no so relevant considering that:

- the fact than SAMe may modify gene expression doesn’t imply that leads to pathology. Authors doesn’t show the problem in using SAMe.

- Even if a problem occurs, authors are using higher doses than those recommended. And many drugs have adverse effects at higher doses than those recommended.  

- Several studies have point out the antagonist effect of SAMe and VPA, but seems like differences may occur depending on age and dose. But the use of VPA in pregnant woman to counteract SAMe-induced gene modification is too risky, which limits the value of this finding.

  1. Another interesting information concerns VPA exposure, which in other studies increased autism rates. Here, VPA exposure did not cause modification of gene expression shorter after treatment, so other underlying mechanisms linking VPA and ASD should be explored. However, experimental conditions, dose and timing may be different to the studies relating VPA with ASD, so for this information to be interesting it should be clearly stated in the manuscript than experimental conditions are similar to those relating VPA to ASD.

Nonetheless, the study by Weinstein-Fudim et al. suffers from several weaknesses:

  1. The studied brain region is vaguely described and must be more specific: i.e. frontal part of brain, does it mean frontal lobe? A good way to describe the region could be using bregma coordinates, a diagram of the region sampled and a picture. Was dissection accurate and similar in all samplings?

Since it appears imprecise, it would be needed to have a high number of samples to identify variability due to sampling.

However,

  1. The number of samples is low (i.e. 2 samples for males in the SAMe treated group). P-values can therefore no be accurately calculated. No other method was used to validate results (e.g. in situ hybridization for some of the most representative and differentially expressed genes).
  2. Additionally, statistical information may be misleading because authors have used low "Thresholds". For instance, Log2FC are commonly accepted at values <-1.5 and >1.5. However, in table 1, Log2FC in between this range were also considered as significant. In other tables Log2FC is not even described. The use of different FC could be justified by providing a Volcano plot supporting this decision (for instance as supplemental figure).

The full dataset of results (absolute value, FC, Q, p-value, etc. per gene, for all significant genes should be available, at least as supplementary material. For instance, Fig. 1 depicts the number of DEG in females vs. males (170), table 1 only show some selected genes (15) with debatable Log2FC.

  1. Some sentences are based on external studies that do not allow to state the hypotheses developed there. Eg. Section Results 2.1 “As the treated mice were not followed to adulthood since gene expression of the neonates obviously requires sacrifice of animals, we do not know if these specific animals would have developed ASD like behavioral changes. However, many studies have shown that prenatal administration of VPA to mice induced autistic like behavior [4, 6, 8].”

It should be clearly stated that in these many studies the dose and time of administration was the same than here (or not). Authors could use other specimens exposed to similar dose exposures and conditions, and leave them growth to adult life to check ASD traits.

Due to the above observations, I believe it is not recommendable to approve this document for publication, unless substantive changes and additional experiments are made.

The present study seems a good preliminary screen to be further exploited with additional experimental work, that could include:

- adding number of samples in a specific precise and well determined brain region. Justification about why this brain region was studied and no other should be included.

- validation of some of the results: e.g. imaging some the genes differentially expressed in males and females by ISH/RNAscope, showing these differences between sexes also with the images/counting.

- is it realistic to add a sample group for SAMe with similar doses to the used in human?

- It would be useful to provide information on physiological parameters and behavior in adult mice similarly exposed in utero, in order to determine a tentative negative effect of the antioxidant SAMe that sustain the recommendation to pregnant woman (e.g. deepen into the VEGF pathway pointed out in this screen); as well as to demonstrate that VPA effects under these experimental conditions of dose and time on ASD behavioral traits are not related to changes in gene expression shorter after treatment exposure were not observed. This information would complement previous studies according to last paragraph in section 3.1.

If editors decide to publish, I would recommend the authors to improve presentation considering all the above mentioned comments. In addition, authors may want:

- bigger heatmaps,

- shorter text and more focused, especially in the discussion since sometimes it seems like a review with no linkage with the results obtained here. Other text may also be reduced such as section 4.2. 

- Place properly arrows in table 4.

- Fig. 4, please add label (i.e. symbol gene) to the points with higher or lower Fold change, to make more visual what are the most changing genes.

- Provide graph and SEM for Body Weight: ‘SAMe administration did not change the litter size but slightly reduced the weight of the offspring (1.48g vs 1.74 in the controls, P=0.10).’

- For the dose of SAMs recommended in human by body weight: you stated ‘one third the dose we used’. Please, provide also numbers referred to body weight.

- Add number of reference (e.g. Intro: Choi et al… [?]).

- To check minor spelling:

e.g. Abstract: One of the more (most?).

e.g Section 3.2: is dynamically changes (changed?).

- To delete all double spaces in between words (e.g. Intro: by concomitant       SAMe)

Author Response

Response to reviewer's comments:

We thank the reviewers for their instructive comments. Here is our response.

Reviewer 2:

We thank this reviewer for his comments which we feel will improve the quality of this manuscript.  However, there are several comments which we could not address because they require to carry- out additional experiments and in the current times we are unable to do that. Hence, this is our reply.

  1. The reviewer is right by pointing out that we used a dose that is 3 times higher than the human dose per body weight, as we also stated in the discussion. The usual human dose is about 400-1200 mg daily, namely 8-20 mg/Kg. Our dose is 2-3 times higher than the human dose. Human treatment is daily over long periods of time. We indeed gave a 2-3 times higher dose, but only for 3 days. This issue is now highlighted in the introduction (page 3) and in the discussion (page 20) and in the methods (page 20).

We do not feel that we have sufficient data to claim that SAMe is teratogenic, but only say that we should be more careful and further study this issue. In fact, as mentioned in our reply to reviewer 3, we followed up the offspring for one month and found that weight gain and viability are similar to controls, implying, as this reviewer points out, that if there is a long-term pathology, it is probably minimal. This was added to the discussion in section 3.2 at the end of page 17.

  1. The reviewer is right regarding the teratogenicity of VPA. Although being mentioned earlier we also added in the introduction and discussion a sentence stating that administration of VPA on day 12 of gestation in mice will cause ASD like behavior but not malformations since it is post the organogenetic period for most organs, especially in relation to neural tube closure.
  2. We added in the methods (section 4.3) more details on the parts of the brain studied and at the end of the introduction why this part was chosen. The introduction says: "We focused on the frontal half of the brain of PND1 mice because this part contains both the prefrontal cortex which was found to be affected by postnatal VPA administration in our previous study, and the hippocampus in which neuronal size abnormalities were found in ASD patients [43], Furthermore, structural and genetic changes were also reported in these regions in mice with autistic like behavior [44, 45]".

In the methods we wrote: "A coronal cut was done in the brain, while looking at the upper surface of the cerebrum and cerebellum, exactly in the middle, removing the frontal half of the brain for our studies. Dissection was similar in all specimens and was done by the same person (LW)".

  1. We are aware that the number of male samples is small and clearly wrote that in the manuscript. However, since the data on both pups was very close, our statistician suggested adding this group to the study. We also stated that this is a weakness of this study. Unfortunately, as we do not have more tissues, we cannot carry out additional studies on more specimens.
  2. As written by the reviewer: "Additionally, statistical information may be misleading….The use of different FC could be justified by providing a Volcano plot supporting this decision (for instance as supplemental figure)".

Our response is as follows:  We already included in the main manuscript a ma plot and now added the Volkano plot as supplemental material as suggested by the reviewer.  We also added the full dataset (complete least of genes changed by SAMe) as a complementary material.

  1. We indeed used similar doses of VPA to those used by other investigators. This is now stated. The following sentence is now written: "many studies have shown that prenatal administration of VPA to mice in similar doses to ours and on similar gestational days induced autistic like behavior."
  2. The additional studies suggested by the reviewer are indeed important. Some of them are being carried out at present (i.e. long term postnatal administration of SAMe to VPA treated mice) and others will be carried out in the future but we feel that the data reported here are sufficiently important to be published separately.
  3. We enlarged the heatmaps, slightly reduced the discussion, especially the parts related to the effects of VPA, added changes in section 4; added data on postnatal weight gain including a table and figures and made all other corrections suggested by the reviewer regarding figures, text of legend to figures ext. We also shortened the discussion especially section 3.2 and 3.3, pages 17-20.
  4. Place properly arrows in table 4.

Done

  1. Fig. 4, please add label (i.e. symbol gene) to the points with higher or lower Fold change, to make more visual what are the most changing genes.
  2. We added labels to the higher and lower fold change genes.
  3. Provide graph and SEM for Body Weight: ‘SAMe administration did not change the litter size but slightly reduced the weight of the offspring (1.48g vs 1.74 in the controls, P=0.10).’

Graphs were added as figure 1 A-C

  1. For the dose of SAMe recommended in human by body weight: you stated ‘one third the dose we used’. Please, provide also numbers referred to body weight.

We added the information on page 3 line 10.

  1. Add number of reference (e.g. Intro: Choi et al… [?]).

The reference was added.

  1. To check minor spelling:

Done

  1. Abstract: One of the more (most?).

The word more was changed to most

17 Section 3.2: is dynamically changes (changed?).

The word changes was changed to changed

  1. To delete all double spaces in between words (e.g. Intro: by concomitant SAMe)

We deleted all the double spaces.

Reviewer 3 Report

In this manuscript, the authors conducted a systematic comparison of gene expression in neonates prenatally exposed to VPA, SAMe or both together, and found SAMe could induce sex-specific changes in gene expressions, which were prevented by co-administration by VPA. This study has important implications in guiding maternal diet, therapeutic window in time and dose for SAMe for treating ASD.

The following concerns are raised:

  1. In the results section, it indicates SAMe administration slightly reduced the weight of the offspring, does this effect display gender-specific, dose this effect disappear at late stages, such as in 2-month-old mice? The raw data of neonates regarding the toxicity of VPA or SAMe or combined administration should be provided. Also gene expression data between VPA and control group should be included.
  2. In the result section 2.1, “However, many studies have shown that prenatal administration of VPA to mice induced autistic-like behavior [4, 6, 8]”.  Does the reference 8 show prenatal administration?
  3. The references 8 and 16 are the same.
  4. All main figures should be well organized or include more details in legends. For instance, figure 2 should be reframed to concisely display the key points.
  5. It may explain why the frontal part of the brain is focused and examined in this study.
  6. In the discussion section, the authors should include more interpretation for SAMe-induced genes in the context of neurophysiology and epigenetic teratogens, or the possible reason for sex-specific effects.

Author Response

Response to reviewer's comments:

We thank the reviewers for their instructive comments. Here is our response.

Reviewer 3:  

We thank this reviewer for his comments which we feel will improve the quality of this manuscript. 

  1. Relating to the comment on neonatal weight and postnatal weight gain: This is a very important comment. We indeed followed the weight gain of the offspring to the age of one month, and added this data to the results with a table and figures. There was no difference in weight gain during the first postnatal month. There was also no mortality, implying that there were apparently no serious malformations induced by SAMe. This is now added to the methods and results.
  2. The reviewer is right regarding reference 8, which was removed as a citation in this sentence. Reference 8 is our previous study which evaluated the effects of postnatal administration of SAMe and VPA. We added, however, 2 additional references on this issue.
  3. Reference 16 was removed as it is indeed the same as reference 8.
  4. The figures were corrected, marked by arrows and more details were added in the legends.
  5. We explained in the text why we studied the frontal part of the brain: this was done for 2 main reasons; to compare the data on the newborn mice to our previous data in 60 day old mice and also because many investigators have shown changes in the frontal cortex in different models of ASD like behavior. Moreover, changes were also described in the frontal cortex of children with ASD. We also elaborated according to a comment by reviewer 2 which parts were removed i.e.: "A coronal cut was done in the brain, while looking at the upper surface of the cerebrum and cerebellum, exactly in the middle, removing the frontal half of the brain for our studies". This is written in the methods and in the introduction.
  6. Regarding the comment to include more details in the discussion on the neurophysiology and possible explanation of sex-specific effects of SAMe, we added some data to the discussion (i.e. second paragraph in section 3.2) regarding possible explanation why it affects gene expression if administered prenatally and not postnatally). In addition, we discussed some issues related to the sex specific effects of SAMe (section 3.4), but we should remember that there is very little data on these issues.
  7. We have to point out that following the request of reviewer 2, we made several additional changes in the discussion, especially omitted some paragraphs.

Round 2

Reviewer 2 Report

Authors have addressed most of the minor comments. However, some of the major ones remain unaddressed, including validation of the results (e.g. by ISH/RNAscope of representative genes), increasing number of samples, or demonstration that SAMe may lead to pathology (the fact than SAMe may modify gene expression doesn’t imply anything by itself).

This study seems just a modest preliminary screen. Ideally it should be published with more data and conclusions.

Author Response

We thank again this reviewer for his comments,

  1. Authors have addressed most of the minor comments. However, some of the major ones remain unaddressed, including validation of the results (e.g. by ISH/RNAscope of representative genes), increasing number of samples, or demonstration that SAMe may lead to pathology (the fact than SAMe may modify gene expression doesn’t imply anything by itself).
    This study seems just a modest preliminary screen. Ideally it should be published with more data and conclusions.
  2. As suggested by the academic editor we added a paragraph entitled “Limitation of the study” on page 20 as section 3.5 which describes the relative weakness of the study due to some of the concerns raised by reviewer 2:

Limitations of the study:

This study also has several limitations. First, we have only two male offspring that were prenatally exposed to SAMe. They were included because the data of both animals was very close. We also used doses of SAMe that are 2-3 times higher than the human dose, although the dams were treated for 3 days only as opposed to human treatment that is generally for much longer time. In addition, we do not have any data on gene silencing. Neither did we follow up the offspring to the age when we are able to carry out neurobehavioral studies and evaluate the autistic like behavior induced by prenatal VPA and the possible damaging effects of SAMe. All these issue await additional studies in the future.

Reviewer 3 Report

Thanks for addressing all the concerns, one minor point regarding the format: For section 3.4, it seems the last paragraph is different from other paragraphs in terms of the format. Additionally, Figure 1A should be aligned with Figure 1B and 1C, or organized into a panel. 

Author Response

Reviewer 3

We thank again this reviewer for his comments

  1. For section 3.4, it seems the last paragraph is different from other paragraphs in terms of the format.
  2. We changed the format of the last paragraph in section 3.4

  1. Additionally, Figure 1A should be aligned with Figure 1B and 1C, or organized into a panel.
  2. We organized figure 1A-C into a panel as suggested.
